# Adaptive Methods for Nonconvex Optimization

**Manzil Zaheer** *
Google Research
manzilzaheer@google.com

**Sashank J. Reddi** *
Google Research
sashank@google.com

**Devendra Sachan**
Carnegie Mellon University
dsachan@andrew.cmu.edu

**Satyen Kale**
Google Research
satyenkale@google.com

**Sanjiv Kumar**
Google Research
sanjivk@google.com

## Abstract

Adaptive gradient methods that rely on scaling gradients down by the square root of exponential moving averages of past squared gradients, such RMSPROP, ADAM, ADADELTA have found wide application in optimizing the nonconvex problems that arise in deep learning. However, it has been recently demonstrated that such methods can fail to converge even in simple convex optimization settings. In this work, we provide a new analysis of such methods applied to *nonconvex* stochastic optimization problems, characterizing the effect of increasing minibatch size. Our analysis shows that under this scenario such methods do converge to stationarity up to the statistical limit of variance in the stochastic gradients (scaled by a constant factor). In particular, our result implies that increasing minibatch sizes enables convergence, thus providing a way to circumvent the nonconvergence issues. Furthermore, we provide a new adaptive optimization algorithm, YOGI, which controls the increase in effective learning rate, leading to even better performance with similar theoretical guarantees on convergence. Extensive experiments show that YOGI with very little hyperparameter tuning outperforms methods such as ADAM in several challenging machine learning tasks.

## 1 Introduction

We study nonconvex stochastic optimization problems of the form

$$\min_{x\in\mathbb{R}^d} f(x) := \mathbb{E}_{s\sim\mathbb{P}}[\ell(x,s)], \tag{1}$$

where $\ell$ is a smooth (possibly nonconvex) function and $\mathbb{P}$ is a probability distribution on the domain $\mathcal{S} \subset \mathbb{R}^k$. Optimization problems of this form arise naturally in machine learning where $x$ are model parameters, $\ell$ is the loss function and $\mathbb{P}$ is an unknown data distribution. Stochastic gradient descent (SGD) is the dominant method for solving such optimization problems, especially in nonconvex settings. SGD [29] iteratively updates the parameters of the model by moving them in the direction of the negative gradient computed on a minibatch scaled by step length, typically referred to as learning rate. One has to decay this learning rate as the algorithm proceeds in order to control the variance in the stochastic gradients computed over a minibatch and thereby, ensure convergence. Hand tuning the learning rate decay in SGD is often painstakingly hard. To tackle this issue, several methods that automatically decay the learning rate have been proposed. The first prominent algorithms in this line of research is ADAGRAD [7, 22], which uses a per-dimension learning rate based on squared past gradients. ADAGRAD achieved significant performance gains in comparison to SGD when the gradients are sparse.

Although ADAGRAD has been demonstrated to work well in sparse settings, it has been observed that its performance, unfortunately, degrades in dense and nonconvex settings. This degraded performance

---

is often attributed to the rapid decay in the learning rate when gradients are dense, which is often the case in many machine learning applications. Several methods have been proposed in the deep learning literature to alleviate this issue. One such popular approach is to use gradients scaled down by square roots of exponential moving averages of squared past gradients instead of cumulative sum of squared gradients in ADAGRAD. The basic intuition behind these approaches is to adaptively tune the learning rate based on only the recent gradients; thereby, limiting the reliance of the update on only the past few gradients. RMSPROP, ADAM, ADADELTA are just few of many methods based on this update mechanism [34, 16, 40].

Exponential moving average (EMA) based adaptive methods are very popular in the deep learning community. These methods have been successfully employed in plethora of applications. ADAM and RMSPROP, in particular, have been instrumental in achieving state-of-the-art results in many applications. At the same time, there have also been concerns about their convergence and generalization properties, indicating that despite their widespread use, our understanding of these algorithms is still very limited. Recently, [25] showed that EMA based adaptive methods may not converge to the optimal solution even in simple convex settings when a constant minibatch size is used. Their analysis relies on the fact that the *effective* learning rate (i.e. the learning rate parameter divided by square root of an exponential moving average of squared past gradients) of EMA methods can potentially increase over time in a fairly quick manner, and for convergence it is important to have the learning rate *decrease* over iterations, or at least have controlled increase. This issue persists even if the learning rate parameter is decreased over iterations.

Despite the problem of non-convergence demonstrated by [25], their work does not rule out convergence in case the minibatch size is increased with time, thereby decreasing the variance of the stochastic gradients. Increasing minibatch size has been shown to help convergence in a few optimization algorithms that are not based on EMA methods [10, 3].

**Contributions.** In the light of this background, we state the main contributions of our paper.

- We develop convergence analysis for ADAM under certain useful parameter settings, showing convergence to a stationary point up to the statistical limit of variance in the stochastic gradients (scaled by a constant factor) even for nonconvex problems. Our analysis implies that increasing batch size will lead to convergence, as increasing batch size decreases variance linearly. Our work thus provides a principled means to circumvent the non-convergence results of [25].

- Inspired by our analysis of ADAM and the intuition that controlled increase of effective learning rate is essential for good convergence, we also propose a new algorithm (YOGI) for achieving adaptivity in SGD. Similar to the results in ADAM, we show convergence results with increasing minibatch size. Our analysis also highlights the interplay between level of "adaptivity" and convergence of the algorithm.

- We provide extensive empirical experiments for YOGI and show that it performs better than ADAM in many state-of-the-art machine learning models. We also demonstrate that YOGI achieves similar, or better, results to best performance reported on these models without much hyperparameter tuning.

**Related Work.** The literature in stochastic optimization is vast; so we summarize a few very closely related works. SGD and its accelerated variants for smooth nonconvex problems are analyzed in [8]. Stochastic methods have also been employed in nonconvex finite-sum problems where stronger results can be shown [41, 24, 26, 27]. However, none of these methods are adaptive and can exhibit poor performance in ill-conditioned problems. All the aforementioned works show convergence to a stationary point. Recently, several first-order and second-order methods have been proposed that are guaranteed to converge to local minima under certain conditions [1, 4, 14, 28]. However, these methods are computationally expensive or exhibit slow convergence in practice, making them unsuitable for large-scale settings. Adaptive methods, that include ADAGRAD, RMSPROP, ADAM, ADADELTA have been mostly studied in the convex setting but their analysis in the nonconvex setting is largely missing [7, 22, 40, 34, 16]. Normalized variants of SGD have been studied recently in nonconvex settings [10, 3].

**Notation** For any vectors $a, b \in \mathbb{R}^d$, we use $\sqrt{a}$ for element-wise square root, $a^2$ for element-wise square, $a/b$ to denote element-wise division. For any vector $\theta_i \in \mathbb{R}^d$, either $\theta_{i,j}$ or $[\theta_i]_j$ are used to denote its $j^{\text{th}}$ coordinate where $j \in [d]$.

## 2 Preliminaries

We now formally state the definitions and assumptions used in this paper. We assume function $\ell$ is *L-smooth*, i.e., there exists a constant $L$ such that

$$\|\nabla \ell(x, s) - \nabla \ell(y, s)\| \leq L\|x - y\|, \quad \forall\, x, y \in \mathbb{R}^d \text{ and } s \in \mathcal{S}.$$

Furthermore, also assume that the function $\ell$ has bounded gradient i.e., $\|\nabla[\ell(x, s)]_i\| \leq G$ for all $x \in \mathbb{R}^d$, $s \in \mathcal{S}$ and $i \in [d]$. Note that these assumptions trivially imply that expected loss $f$ defined in (1) is $L$-smooth, i.e., $\|\nabla f(x) - \nabla f(y)\| \leq L\|x - y\|$ for all $x, y \in \mathbb{R}^d$. We also assume the following bound on the variance in stochastic gradients: $\mathbb{E}\|\nabla \ell(x, s) - \nabla f(x)\|^2 \leq \sigma^2$ for all $x \in \mathbb{R}^d$. Such assumptions are typical in the analysis of stochastic first-order methods (cf. [8, 9]).

We analyze convergence rates of some popular adaptive methods for the above classes of functions. Following several previous works on nonconvex optimization [23, 8], we use $\|\nabla f(x)\|^2 \leq \delta$ to measure the "stationarity" of the iterate $x$; we refer to such a solution as $\delta$-accurate solution[2]. In contrast, algorithms in the convex setting are typically analyzed with the suboptimality gap, $f(x) - f(x^*)$, where $x^*$ is an optimal point, as the convergence criterion. However, it is not possible to provide meaningful guarantees for such criteria for general nonconvex problems due to the hardness of the problem. We also note that adaptive methods have historically been studied in online convex optimization framework where the notion of regret is used as a measure of convergence. This naturally gives convergence rates for stochastic convex setting too. In this work, we focus on the stochastic nonconvex optimization setting since that is precisely the right model for risk minimization in machine learning problems.

To simplify the exposition of results in the paper, we define the following measure of efficiency for a stochastic optimization algorithm:

**Definition 1.** *Stochastic first-order (SFO) complexity of an algorithm is defined as the number of gradients evaluations of the function $\ell$ with respect to its first argument made by the algorithm.*

Since our paper only deals with first order methods, the efficiency of the algorithms can be measured in terms of SFO complexity to achieve a $\delta$-accurate solution. Throughout this paper, we hide the dependence of SFO complexity on $L$, $G$, $\|x^0 - x^*\|^2$ and $f(x^0) - f(x^*)$ for a clean comparison. Stochastic gradient descent (SGD) is one of the simplest algorithms for solving (1). The update at the $t^{\text{th}}$ iteration of SGD is of the following form:

$$x_{t+1} = x_t - \eta_t g_t, \tag{SGD}$$

where $g_t = \nabla \ell(x_t, s_t)$ and $s_t$ is a random sample drawn from the distribution $\mathbb{P}$. When the learning rate is decayed as $\eta_t = 1/\sqrt{t}$, one can obtain the following well-known result [8]:

**Corollary 1.** *The SFO complexity of* SGD *to obtain a $\delta$-accurate solution is $O(1/\delta^2)$.*

In practice, it is often tedious to tune the learning rate of SGD because rapid decay in learning rate like $\eta_t = 1/\sqrt{t}$ typically hurts the empirical performance in nonconvex settings. In the next section, we investigate adaptive methods which partially circumvent this issue.

## 3 Algorithms

In this section, we discuss adaptive methods and analyze their convergence behavior in the nonconvex setting. In particular, we focus on two algorithms: ADAM (3.1) and the proposed method, YOGI (3.2).

### 3.1 Adam

ADAM is an adaptive method based on EMA, which is popular among the deep learning community [16]. EMA based adaptive methods were initially inspired from ADAGRAD and were proposed to address the problem of rapid decay of learning rate in ADAGRAD. These methods scale down the gradient by the square roots of EMA of past squared gradients.

The pseudocode for ADAM is provided in Algorithm 1. The terms $m_t$ and $v_t$ in Algorithm 1 are EMA of the gradients and squared gradients respectively. Note that here, for the sake of clarity, the

| **Algorithm 1** ADAM | **Algorithm 2** YOGI |
|---|---|
| **Input:** $x_1 \in \mathbb{R}^d$, learning rate $\{\eta_t\}_{t=1}^T$, decay parameters $0 \leq \beta_1, \beta_2 \leq 1, \epsilon > 0$ | **Input:** $x_1 \in \mathbb{R}^d$, learning rate $\{\eta_t\}_{t=1}^T$, parameters $0 < \beta_1, \beta_2 < 1, \epsilon > 0$ |
| Set $m_0 = 0, v_0 = 0$ | Set $m_0 = 0, v_0 = 0$ |
| **for** $t = 1$ **to** $T$ **do** | **for** $t = 1$ **to** $T$ **do** |
| $\quad$ Draw a sample $s_t$ from $\mathbb{P}$. | $\quad$ Draw a sample $s_t$ from $\mathbb{P}$. |
| $\quad$ Compute $g_t = \nabla\ell(x_t, s_t)$. | $\quad$ Compute $g_t = \nabla\ell(x_t, s_t)$. |
| $\quad m_t = \beta_1 m_{t-1} + (1 - \beta_1)g_t$ | $\quad m_t = \beta_1 m_{t-1} + (1 - \beta_1)g_t$ |
| $\quad v_t = v_{t-1} - (1 - \beta_2)(v_{t-1} - g_t^2)$ | $\quad v_t = v_{t-1} - (1 - \beta_2)\mathrm{sign}(v_{t-1} - g_t^2)g_t^2$ |
| $\quad x_{t+1} = x_t - \eta_t m_t/(\sqrt{v_t} + \epsilon)$ | $\quad x_{t+1} = x_t - \eta_t m_t/(\sqrt{v_t} + \epsilon)$ |
| **end for** | **end for** |

debiasing step used in the original paper by [16] is removed but our results also apply to the debiased version. A value of $\beta_1 = 0.9$, $\beta_2 = 0.999$ and $\epsilon = 10^{-8}$ is typically recommended in practice [16]. The $\epsilon$ parameter, which was initially designed to avoid precision issues in practical implementations, is often overlooked. However, it has been observed that very small $\epsilon$ in some applications has also resulted in performance issues, indicating that it has a role to play in convergence of the algorithm. Intuitively $\epsilon$ captures the amount of "adaptivity" in ADAM: larger values of $\epsilon$ imply weaker adaptivity since $\epsilon$ dominates $v_t$ in this case.

Very recently, [25] has shown non-convergence of ADAM in simple online convex settings, assuming constant minibatch sizes. These results naturally apply to the nonconvex setting too. It is, however, interesting to consider the case of ADAM in nonconvex setting with *increasing batch sizes*.

To this end, we prove the following convergence result for nonconvex setting. In this paper, for the sake of simplicity, we analyze the case where $\beta_1 = 0$, which is typically referred to as RMSPROP. However, our analysis should extend to the general case as well.

**Theorem 1.** *Let $\eta_t = \eta$ for all $t \in [T]$. Furthermore, assume that $\epsilon$, $\beta_2$ and $\eta$ are chosen such that the following conditions satisfied: $\eta \leq \frac{\epsilon}{2L}$ and $1 - \beta_2 \leq \frac{\epsilon^2}{16G^2}$. Then for $x_t$ generated using ADAM (Algorithm 1), we have the following bound*

$$\mathbb{E}\|\nabla f(x_a)\|^2 \leq O\left(\frac{f(x_1) - f(x^*)}{\eta T} + \sigma^2\right),$$

*where $x^*$ is an optimal solution to the problem in* (1) *and $x_a$ is an iterate uniformly randomly chosen from $\{x_1, \cdots, x_T\}$.*

All the proofs are relegated to the Appendix due to spaces constraints. The above result shows that ADAM achieves convergence to stationarity within the constant factor of $O(\sigma^2)$ for constant learning rate $\eta$, which is similar to the result for SGD with constant learning rate [8]. An immediate consequence of this result is that increasing minibatch size can improve convergence. Specifically, the above result assumes a minibatch size of $1$. Suppose instead we use a minibatch size of $b$, and in each iteration of ADAM we average $b$ stochastic gradients computed at the $b$ samples in the minibatch. Since the samples in the minibatch are independent, the variance of the averaged stochastic gradient is at most $\frac{\sigma^2}{b}$, a factor $b$ lower than a single stochastic gradient. Plugging this variance bound into the bound of Theorem 1, we conclude that increasing the minibatch size decreases the limiting expected stationarity by a factor of $b$. Specifically, we have the following result which is an immediate consequence of Theorem 1 with fixed batch size $b$ and constant learning rate.

**Corollary 2.** *For $x_t$ generated using ADAM with constant $\eta$ (and parameters from Theorem 1), we have*

$$\mathbb{E}[\|\nabla f(x_a)\|^2] \leq O\left(\frac{1}{T} + \frac{1}{b}\right),$$

*where $x_a$ is an iterate uniformly randomly chosen from $\{x_1, \cdots, x_T\}$.*

The above results shows that ADAM obtains a point that has bounded stationarity in expectation i.e., $\mathbb{E}[\|\nabla f(x_a)\|^2] \leq O(1/b)$ as $T \to \infty$. Note that this does not necessarily imply that the $x_a$ is close to a stationary point but a small bound is typically sufficient for many machine learning applications. To ensure good SFO complexity, we need $b = \Theta(T)$, which yields the following important corollary.

**Corollary 3.** ADAM *with $b = \Theta(T)$ and constant $\eta$ (and parameters from Theorem 1), we obtain* $\mathbb{E}[\|\nabla f(x_a)\|^2] \leq O(1/T)$ *and the SFO complexity for achieving a $\delta$-accurate solution is $O(1/\delta^2)$.*

The result simply follows by using batch size $b = \Theta(T)$ and constant $\eta$ in Theorem 1. Note that this result can be achieved using a constant learning rate and $\beta_2$. We defer further discussion of these bounds to the end of the section.

## 3.2 Yogi

The key element behind ADAM is to use an adaptive gradient while ensuring that the learning rate does not decay quickly. To achieve this, ADAM uses an EMA which is, by nature, multiplicative. This leads to a situation where the past gradients are forgotten in a fairly fast manner. As pointed out in [25], this can especially be problematic in sparse settings where gradients are rarely nonzero. An alternate approach to attain the same goal as ADAM is through additive updates. To this end, we propose a simple additive adaptive method, YOGI, for optimizing the stochastic nonconvex optimization problem of our interest. (Name derived from the Sanskrit word yuj meaning to add.)

Algorithm 2 provides the pseudocode for YOGI. Note that the update looks very similar to ADAGRAD except for the use of $\text{sign}(v_{t-1} - g_t^2)$ in YOGI. Similar to ADAM, $\epsilon$ controls the amount of adaptivity in the method. The difference with ADAM is in the update of $v_t$. To gain more intuition for YOGI, let us compare its update rule with that of ADAM. The quantity $v_t - v_{t-1}$ is $-(1 - \beta_2) \, \text{sign} \, (v_{t-1} - g_t^2) g_t^2$ in YOGI as opposed to $-(1 - \beta_2)(v_{t-1} - g_t^2)$ in ADAM. An important property of YOGI, which is common with ADAM, is that the difference of $v_t$ and $v_{t-1}$ depends only on $v_{t-1}$ and $g_t^2$. However, unlike ADAM, the magnitude of this difference in YOGI only depends on $g_t^2$ as opposed to dependence on both $v_{t-1}$ and $g_t^2$ in ADAM. Note that when $v_{t-1}$ is much larger than $g_t^2$, ADAM and YOGI increase the effective learning rate. However, in this case it can be seen that ADAM can rapidly increase the effective learning rate while YOGI does it in a controlled fashion. As we shall see shortly, we often observed improved empirical performance by adopting such a controlled increase in effective learning rate. Even in cases where rapid change in learning rate is desired, one can use YOGI with a smaller value of $\beta_2$ to mirror that behavior. Also, note that YOGI has same $O(d)$ computational and memory requirements as ADAM, and is hence, efficient to implement.

Similar to ADAM, we provide the following convergence result for YOGI in the nonconvex setting.

**Theorem 2.** *Let $\eta_t = \eta$ for all $t \in [T]$. Furthermore, assume that $\epsilon$, $\beta_2$ and $\eta$ are chosen such that the following conditions satisfied: $1 - \beta_2 \leq \frac{\epsilon^2}{16G^2}$ and $\eta \leq \frac{\epsilon \sqrt{\beta_2}}{2L}$. Then for $x_t$ generated using YOGI (Algorithm 2), we have the following bound*

$$\mathbb{E}\|\nabla f(x_a)\|^2 \leq O\left(\frac{f(x_1) - f(x^*)}{\eta T} + \sigma^2\right),$$

*where $x^*$ is an optimal solution to the problem in (1) and $x_a$ is an iterate uniformly randomly chosen from $\{x_1, \cdots, x_T\}$.*

The convergence result is very similar to the result in Theorem 1. As before, the following results on bounded gradient norm with increasing batch size can be obtained as a simple corollary of Theorem 2.

**Corollary 4.** *For $x_t$ generated using YOGI with constant $\eta$ (and parameters from Theorem 2), we have*

$$\mathbb{E}[\|\nabla f(x_a)\|^2] \leq O\left(\frac{1}{T} + \frac{1}{b}\right)$$

*where $x_a$ is an iterate uniformly randomly chosen from $\{x_1, \cdots, x_T\}$.*

**Corollary 5.** *YOGI with $b = \Theta(T)$ and constant $\eta$ (and parameters from Theorem 2) has SFO complexity is $O(1/\delta^2)$ for achieving a $\delta$-accurate solution.*

**Discussion about Theoretical Results.**  We would like to emphasize that the SFO complexity obtained here for ADAM or YOGI with large batch size is similar to that of SGD (see Corollary 1). While we stated our theoretical results with batch size $b = \Theta(T)$ for the sake of simplicity, similar results can be obtained for increasing minibatches $b_t = \Theta(t)$. In practice, we believe a much weaker increase in batch size is sufficient. In fact, when the variance is not large, our analysis shows that a reasonably large batch size can work well. Before we proceed further, note that these are upper bounds and may not be completely reflective of the performance in practice. It is, however, instructive to note the relationship between different quantities of these algorithms in our results. In particular, the amount of adaptivity that can be tolerated depends on the parameter $\beta_2$. This convergence analysis

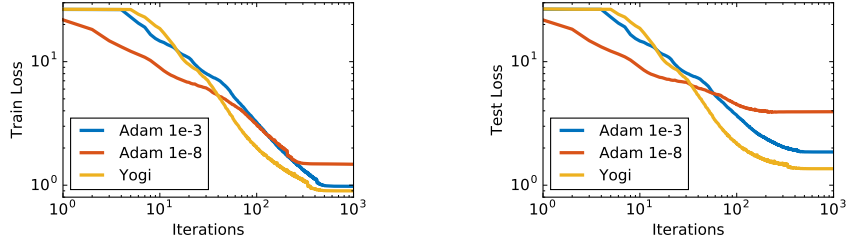

Figure 1: Effect of adaptivity level $\epsilon$ on training MNIST deep autoencoder. We observe aggressive adaptivity leads to poor performance while a more controlled variation like in YOGI leads to a much better final performance.

is useful when $\frac{\epsilon}{G}$ is large when compared to $1 - \beta_2$ i.e., the adaptivity level is moderate[3]. Recall that $\epsilon$ here is only a parameter of the algorithm and is not associated with accuracy of the solution. Typically, it is often desirable to have small $\epsilon$ in adaptive methods; however, as we shall see in the experiment section, limiting the adaptivity level to a certain extent almost always improves the performance (e.g. see Table 1 and 2, and Figure 1). For this reason, we also set the adaptivity level to a moderate value of $\epsilon = 10^{-3}$ for YOGI across all our experiments.

## 4 Experiments

Based on the insights gained from our theoretical analysis, we now present empirical results showcasing three aspects of our framework: (i) the value gained by controlled variation in learning rate using YOGI, (ii) fast optimization, and (iii) wide applicability on several large-scale problems ranging from natural language processing to computer vision. We run our experiments on a commodity machine with Intel® Xeon® CPU E5-2630 v4 CPU, 256GB RAM, and 8 Nvidia® Titan Xp GPU.

**Experimental Setup.** We compare YOGI with highly tuned ADAM and the reported state-of-the-art results for the setup. Typically, for obtaining the state-of-the-art results extensive hyper-parameter tuning and carefully designed learning rate schedules are required (see e.g. Transformers [35][Equation 8]). However, for YOGI, we restrict to tuning just two scalar hyperparameters — the learning rate and $\beta_2$. In all our experiments, YOGI and ADAM are always initialized from the same point. Initialization of $m_t$ and $v_t$ are also important for YOGI and ADAM. These are often initialized with 0 in conjunction with debiasing strategies [16]. Even with additional debiasing, such an initialization can result in very large learning rates at start of the training procedure; thereby, leading to convergence and performance issues. Thus, for YOGI, we propose to initialize the $v_t$ based on gradient square evaluated at the initial point averaged over a (reasonably large) mini-batch. Decreasing learning rate is typically necessary for superior performance. To this end, we chose the simple learning rate schedule of reducing the learning rate by a constant factor when performance metric plateaus on the validation/test set (commonly known as ReduceLRonPlateau). Inspired from our theoretical analysis, we set a moderate value of $\epsilon = 10^{-3}$ in YOGI for all the experiments in order to control the adaptivity. We will see that under such simple setup, YOGI still achieves similar or better performance compared to highly task-specific tuned setups. Due to space constraints, a few experiments are relegated to the Appendix (Section D).

**Deep AutoEncoder.** For our first task, we train two deep autoencoder models from [12] called "CURVES" and "MNIST", typically used as standard benchmarks for neural network optimization (see e.g. [21, 32, 36, 20]). The "MNIST" autoencoder consists of an encoder with layers of size $(28 \times 28)$-1000-500-250-30 and a symmetric decoder, totaling in 2.8M parameters. The thirty units in the code layer were linear and all the other units were logistic. The data set contains images of handwritten digits 0–9. The pixel intensities were normalized to lie between 0 and 1. The "CURVES" autoencoder consists of an encoder with layers of size $(28 \times 28)$-400-200-100- 50-25-6 and a symmetric decoder totaling in 0.85M parameters. The six units in the code layer were linear and all the other units were logistic. In "MNIST" autoencoder, we perform significantly better than all prior results including ADAM with specially tuned learning rate. We also observed similar gains in "CURVES" autoencoder with a smaller $\beta_2$.

Table 1: Train and test loss comparison for Deep AutoEncoders. Standard errors with $2\sigma$ are shown over 6 runs are shown. All our experiments were run for 5000 epochs utilizing the ReduceLRonPlateau schedule with patience of 20 epochs and decay factor of 0.5 with a batch size of 128.

| Method | LR | $\beta_1$ | $\beta_2$ | $\epsilon$ | MNIST | |
| --- | --- | --- | --- | --- | --- | --- |
| | | | | | Train Loss | Test Loss |
| PT + NCG [20] | - | - | - | - | 2.31 | 2.72 |
| RAND+HF [20] | - | - | - | - | 1.64 | 2.78 |
| PT + HF [20] | - | - | - | - | 1.63 | 2.46 |
| KSD [36] | - | - | - | - | 1.8 | 2.5 |
| HF [36] | - | - | - | - | 1.7 | 2.7 |
| ADAM (Default) | $10^{-3}$ | 0.9 | 0.999 | $10^{-8}$ | $1.85 \pm 0.19$ | $4.36 \pm 0.33$ |
| ADAM (Modified) | $10^{-3}$ | 0.9 | 0.999 | $10^{-3}$ | $0.91 \pm 0.04$ | $1.88 \pm 0.07$ |
| YOGI (Ours) | $10^{-2}$ | 0.9 | 0.9 | $10^{-3}$ | $0.78 \pm 0.02$ | $1.70 \pm 0.03$ |
| YOGI (Ours) | $10^{-2}$ | 0.9 | 0.999 | $10^{-3}$ | $0.88 \pm 0.02$ | $\mathbf{1.36 \pm 0.01}$ |

Table 2: Test BLEU score comparison for Base Transformer model [35]. The experiment of En-Vi was run for 30 epochs using a batch size of 3K words/batch for source and target sentences, while En-De was run for 20 epochs using a batch size of 32K words/batch for source and target sentences. We utilized the ReduceLRonPlateau schedule with a patience of 5 epochs and a decay factor of 0.7. The results reported here for our experiments are without checkpoint averaging; Adam+Noam: refers to the special custom learning rate schedule used in [35]; Avg: refers to checkpoint averaging used for the reporting of En-De BLEU scores.

| Method | LR | $\beta_1$ | $\beta_2$ | $\epsilon$ | BLEU | |
| --- | --- | --- | --- | --- | --- | --- |
| | | | | | En-Vi | En-De |
| Adam+Noam+Avg [35] | - | - | - | - | - | 27.3 |
| Adam+Noam (tensor2tensor) [30] | - | - | - | - | 28.1 | |
| SGD+Mom [6] | - | - | - | - | 28.9 | - |
| ADAM (Default) | $10^{-4}$ | 0.9 | 0.999 | $10^{-8}$ | $27.92 \pm 0.22$ | - |
| ADAM (Modified) | $10^{-4}$ | 0.9 | 0.999 | $10^{-3}$ | $28.28 \pm 0.29$ | - |
| ADAM+Noam | $-$ | 0.9 | 0.997 | $10^{-9}$ | $28.84 \pm 0.24$ | - |
| YOGI (Ours) | $10^{-3}$ | 0.9 | 0.99 | $10^{-3}$ | $\mathbf{29.27 \pm 0.07}$ | 27.2 |

**Neural Machine Translation.** As a large-scale experiment, we use the Transformer (TF) model [35] for machine translation, which has recently gained a lot of attention. In TF, both encoder and decoder consists of only self-attention and feed-forward layers. TF networks are known to be notoriously hard to optimize. The original paper proposed a method based on linearly increasing the learning rate for a specified number of optimization steps followed by inverse square root decay. In our experiments, we use the same 6-layer 8-head TF network described in the original paper: The position-wise feed-forward networks have 512 dimensional input/output with a 2048 dimensional hidden layer and ReLU activation. Layer normalization [2] is applied at the input and residual connections are added at the output of each sublayer. Word embeddings between encoder and decoder are shared and the softmax layers are tied. We perform experiments on the IWSLT'15 En-Vi [18] and WMT'14 En-De datasets with the standard train, validation and test splits. These datasets consist of 133K and 4.5M sentences respectively. In both the experiments, we encode the sentences using 32K merge operations using byte-pair encoding [31]. Due to very large-scale nature of the En-De dataset, we only report the performance of YOGI on it. As seen in Table 2, with very little parameter tuning, YOGI is able to obtain much better BLEU score over previously reported results on En-Vi dataset and is directly competitive on En-De, without using any ensembling techniques such as checkpoint averaging.

**ResNets and DenseNets.** For our next experiment, we use YOGI to train ResNets [11] and DenseNets [13], which are very popular architectures, producing state-of-the-art results across many computer vision tasks. Training these networks typically requires careful selection of learning rates. It is widely believed that adaptive methods yield inferior performance for these type of networks

Table 3: Test accuracy for ResNets on CIFAR-10. Standard errors with $2\sigma$ over 6 runs are shown. All our experiments were run for 500 epochs utilizing the ReduceLRonPlateau schedule with patience of 20 epochs and decay factor of 0.5 with a batch size of 128. Also we report numbers from original paper for reference, which employs a highly tuned learning rate schedule.

| Method | LR | $\beta_1$ | $\beta_2$ | $\epsilon$ | Test Accuracy (%) | |
| --- | --- | --- | --- | --- | --- | --- |
| | | | | | ResNet20 | ResNet50 |
| SGD+Mom [11] | - | - | - | - | 91.25 | 93.03 |
| ADAM (Default) | $10^{-3}$ | 0.9 | 0.999 | $10^{-8}$ | $90.37 \pm 0.24$ | $92.59 \pm 0.23$ |
| ADAM (Default) | $10^{-2}$ | 0.9 | 0.999 | $10^{-8}$ | $89.11 \pm 0.22$ | $88.82 \pm 0.33$ |
| ADAM (Modified) | $10^{-3}$ | 0.9 | 0.999 | $10^{-3}$ | $89.99 \pm 0.30$ | $91.74 \pm 0.33$ |
| ADAM (Modified) | $10^{-2}$ | 0.9 | 0.999 | $10^{-3}$ | $92.56 \pm 0.14$ | $93.42 \pm 0.16$ |
| YOGI (Ours) | $10^{-2}$ | 0.9 | 0.999 | $10^{-3}$ | $\mathbf{92.62 \pm 0.17}$ | $\mathbf{93.90 \pm 0.21}$ |

Table 4: Test accuracy for DenseNet on CIFAR10. Standard errors with $2\sigma$ are shown over 6 runs are shown. All our experiments were run for 300 epochs utilizing the ReduceLRonPlateau schedule with patience of 20 epochs and decay factor of 0.5 with a batch size of 64. Also we report numbers from original paper for reference, which employs a highly tuned learning rate schedule.

| Method | LR | $\beta_1$ | $\beta_2$ | $\epsilon$ | Test Accuracy (%) |
| --- | --- | --- | --- | --- | --- |
| SGD+Mom [13] | - | - | - | - | 94.76 |
| ADAM (Default) | $10^{-3}$ | 0.9 | 0.999 | $10^{-8}$ | $92.53 \pm 0.20$ |
| ADAM (Modified) | $10^{-3}$ | 0.9 | 0.999 | $10^{-3}$ | $93.35 \pm 0.21$ |
| YOGI (Ours) | $10^{-2}$ | 0.9 | 0.999 | $10^{-3}$ | $\mathbf{94.38 \pm 0.26}$ |

[15]. We attempt to tackle this challenging problem on the CIFAR-10 dataset. For ResNet experiment, we select a small ResNet network with 20 layers and medium-sized ResNet network with 50 layers (same as those used in original ResNet paper [11] for CIFAR-10). For DenseNet experiment, we used a DenseNet with 40 layers and growth rate $k = 12$ without bottleneck, channel reduction, or dropout. We adopt a standard data augmentation scheme (mirroring/shifting) that is widely used. As seen from Table 3 and 4, without any tuning, our default parameter setting achieves state-of-the-art results for these networks.

**DeepSets.** We also evaluate our approach on the task of classification of point clouds. We use DeepSets [39] to classify point-cloud representation on a benchmark ModelNet40 dataset [38]. We use the same network described in the DeepSets paper: The network consists of 3 permutation-equivariant layers with 256 channels followed by maxpooling over the set structure. The resulting vector representation of the set is then fed to a fully connected layer with 256 units followed by a 40-way softmax unit. We use *tanh* activation at all layers and apply dropout on the layers after set-max-pooling (two dropout operations) with 50% dropout rate. Consistent with our previous experiments, we observe that YOGI outperforms ADAM and obtains better results than those reported in the original paper (Table 6).

**Named Entity Recognition (NER).** Finally, we test YOGI for sequence labeling task in NLP involving recurrent neural networks. We use the popular CNN-LSTM-CRF model [5], [19] for NER task. In this, multiple CNN filters of different widths are used to encode morphological and lexical information observed in characters. A word-level bidirectional LSTM layer models the long-range dependence structure in texts. A linear-chain CRF layer models the sequence-level data likelihood while inference is performed using Viterbi Algorithm. In our experiments, we use the BC5CDR biomedical data [17] [4]. The CNN-LSTM model comprises of 1400 CNN filters of widths [1-7], 300 dimensional pre-trained word embeddings, a single layer 256 dimensional bidirectional LSTM, and dropout probability of 0.5 applied to word embeddings and LSTM output layer. We use exact match to evaluate the F1 score of our approach. The results are presented in Table 7 (Appendix). YOGI, again, performs better than ADAM and achieves better F1 score than the one previously reported.

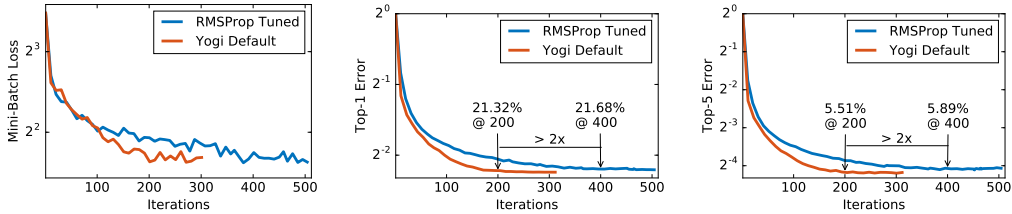

Figure 2: Comparison of highly tuned RMSProp optimizer with YOGI for Inception-Resnet-v2 on Imagenet. First plot shows the mini-batch estimate of the loss during training, while the remaining two plots show top-1 and top-5 error rates on the held-out Imagenet validation set. We observe that YOGI default (learning rate of $10^{-2}$, $\beta_1 = 0.9$, $\beta_2 = 0.999$, $\epsilon = 10^{-3}$) not only achieves better generalization but also reaches similar accuracy level as the tuned optimizer more than 2x faster.

## 5  Discussion

We discuss our observations about empirical results and suggest some practical guidelines for using YOGI. As we have shown, YOGI tends to perform superior to ADAM and highly hand-tuned SGD across various architectures and datasets. It is also interesting to note that typically YOGI has a smaller variance across different runs when compared to ADAM. Furthermore, very little hyperparameter tuning was used for YOGI to remain faithful to our goal. Inspired from our theoretical analysis, we fixed $\epsilon = 10^{-3}$ and $\beta_1 = 0.9$ in all experiments. We also observed that initial learning rate $\eta$ in the range $[10^{-2}, 10^{-3}]$ and $\beta_2 = \{0.9, 0.99, 0.999\}$ appears to work well in most settings. As a general remark, the learning rate for YOGI seems to be around $5 - 10$ times higher than that of ADAM. For decreasing the learning rate, we used the standard heuristic of ReduceLRonPlateau [5] (see Section 4 for more details). In general, the patience of ReduceLRonPlateau scheduler will depend on data size, but in most experiments we saw a patience of 5,000-10,000 gradient steps works well with a decay factor of 0.5.

Finally, we would like to emphasize that other popular learning rate decay schedules different from ReduceLRonPlatueau can also be used for YOGI. Also, it is easy to deploy YOGI in existing pipelines to provide good performance with minimal tuning. To illustrate these points, we conducted an experiment on Inception-Resnet-v2 [33], a high performing model on ImageNet 2012 classification challenge involving $1.28$M images and $1,000$ classes. In order to achieve best results, [33] employed a heavily tuned RMSPROP optimizer with learning rate of $0.045$, momentum decay of $0.9$, $\beta = 0.9$, and $\epsilon = 1$. The learning rate of RMSPROP is decayed every 3 epochs by a factor of $0.94$[6]. With this setting of RMSPROP and batch size of $320$, we obtained a top-1 and top-5 error of $21.63\%$ and $5.79\%$ respectively, under single crop evaluation on the Imagenet validation set comprising of $50K$ images (slightly lower than the published results). However, using YOGI with essentially the default parameters (i.e. learning rate of $10^{-2}$, $\beta_1 = 0.9$, $\beta_2 = 0.999$, $\epsilon = 10^{-3}$) and an alternate decay schedule of reducing learning rate by $0.7$ every 3 epochs achieves better performance (top-1 and top-5 error of $21.14\%$ and $5.46\%$ respectively). Furthermore, it reaches similar accuracy level as the tuned optimizer more than 2x faster, as shown in Figure 2, demonstrating the efficacy of YOGI.

## Footnotes

[2]Here we use $\delta$ instead of standard $\epsilon$ in optimization and machine learning literature since $\epsilon$ symbol is reserved for description of some popular adaptive methods like ADAM.

[3]Note that here, we have assumed same bound $|[\nabla \ell(x, s)]_i| \leq G$ across all coordinates $i \in [d]$ for simplicity, but our analysis can easily incorporate non-uniform bounds on gradients across coordinates.

[4] http://www.biocreative.org/tasks/biocreative-v/track-3-cdr/

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
