[Supplementary Material]

# Appendix

## A    Proof of Theorem 1

We analyze the convergence of ADAM for general minibatch size here. Theorem 1 is obtained by setting $b = 1$. Recall that the update of ADAM is the following

$$x_{t+1,i} = x_{t,i} - \eta_t \frac{g_{t,i}}{\sqrt{v_{t,i}} + \epsilon},$$

for all $i \in [d]$. Since the function $f$ is $L$-smooth, we have the following:

$$f(x_{t+1}) \leq f(x_t) + \langle \nabla f(x_t), x_{t+1} - x_t \rangle + \frac{L}{2} \|x_{t+1} - x_t\|^2$$

$$= f(x_t) - \eta_t \sum_{i=1}^{d} \left( [\nabla f(x_t)]_i \times \frac{g_{t,i}}{\sqrt{v_{t,i}} + \epsilon} \right) + \frac{L\eta_t^2}{2} \sum_{i=1}^{d} \frac{g_{t,i}^2}{(\sqrt{v_{t,i}} + \epsilon)^2} \quad (2)$$

The second step follows simply from ADAM's update. We take the expectation of $f(x_{t+1})$ in the above inequality:

$$\mathbb{E}_t[f(x_{t+1})] \leq f(x_t) - \eta_t \sum_{i=1}^{d} \left( [\nabla f(x_t)]_i \times \mathbb{E}_t \left[ \frac{g_{t,i}}{\sqrt{v_{t,i}} + \epsilon} \right] \right) + \frac{L\eta_t^2}{2} \sum_{i=1}^{d} \mathbb{E}_t \left[ \frac{g_{t,i}^2}{(\sqrt{v_{t,i}} + \epsilon)^2} \right]$$

$$= f(x_t) - \eta_t \sum_{i=1}^{d} \left( [\nabla f(x_t)]_i \times \mathbb{E}_t \left[ \frac{g_{t,i}}{\sqrt{v_{t,i}} + \epsilon} - \frac{g_{t,i}}{\sqrt{\beta_2 v_{t-1,i}} + \epsilon} + \frac{g_{t,i}}{\sqrt{\beta_2 v_{t-1,i}} + \epsilon} \right] \right)$$

$$+ \frac{L\eta_t^2}{2} \sum_{i=1}^{d} \mathbb{E}_t \left[ \frac{g_{t,i}^2}{(\sqrt{v_{t,i}} + \epsilon)^2} \right]$$

$$= f(x_t) - \eta_t \sum_{i=1}^{d} \left( [\nabla f(x_t)]_i \times \left[ \frac{[\nabla f(x_t)]_i}{\sqrt{\beta_2 v_{t-1,i}} + \epsilon} + \mathbb{E}_t \left[ \frac{g_{t,i}}{\sqrt{v_{t,i}} + \epsilon} - \frac{g_{t,i}}{\sqrt{\beta_2 v_{t-1,i}} + \epsilon} \right] \right] \right)$$

$$+ \frac{L\eta_t^2}{2} \sum_{i=1}^{d} \mathbb{E}_t \left[ \frac{g_{t,i}^2}{(\sqrt{v_{t,i}} + \epsilon)^2} \right]$$

$$\leq f(x_t) - \eta_t \sum_{i=1}^{d} \frac{[\nabla f(x_t)]_i^2}{\sqrt{\beta_2 v_{t-1,i}} + \epsilon} + \eta_t \sum_{i=1}^{d} |[\nabla f(x_t)]_i| \left| \mathbb{E}_t \underbrace{\left[ \frac{g_{t,i}}{\sqrt{v_{t,i}} + \epsilon} - \frac{g_{t,i}}{\sqrt{\beta_2 v_{t-1,i}} + \epsilon} \right]}_{T_1} \right|$$

$$+ \frac{L\eta_t^2}{2} \sum_{i=1}^{d} \mathbb{E}_t \left[ \frac{g_{t,i}^2}{(\sqrt{v_{t,i}} + \epsilon)^2} \right] \quad (3)$$

The second equality follows from the fact that $g_t$ is an unbiased estimate of $\nabla f(x_t)$ i.e., $\mathbb{E}[g_t] = \nabla f(x_t)$. This is possible because $v_{t-1,i}$ is independent of $S_t$ sampled at time step $t$. The terms $T_1$ in the above inequality needs to be bounded in order to show convergence. We obtain the following bound on the term $T_1$:

$$T_1 = \frac{g_{t,i}}{\sqrt{v_{t,i}} + \epsilon} - \frac{g_{t,i}}{\sqrt{\beta_2 v_{t-1,i}} + \epsilon}$$

$$\leq |g_{t,i}| \times \left| \frac{1}{\sqrt{v_{t,i}} + \epsilon} - \frac{1}{\sqrt{\beta_2 v_{t-1,i}} + \epsilon} \right|$$

$$= \frac{|g_{t,i}|}{(\sqrt{v_{t,i}} + \epsilon)(\sqrt{\beta_2 v_{t-1,i}} + \epsilon)} \times \left| \frac{v_{t,i} - \beta_2 v_{t-1,i}}{\sqrt{v_{t,i}} + \sqrt{\beta_2 v_{t-1,i}}} \right|$$

$$= \frac{|g_{t,i}|}{(\sqrt{v_{t,i}} + \epsilon)(\sqrt{\beta_2 v_{t-1,i}} + \epsilon)} \times \frac{(1 - \beta_2) g_{t,i}^2}{\sqrt{v_{t,i}} + \sqrt{\beta_2 v_{t-1,i}}}$$

The third equality is due to the definition of $v_{t-1,i}$ and $v_{t,i}$ in ADAM i.e., $v_{t,i} = \beta_2 v_{t-1,i} + (1-\beta_2)g_{t,i}^2$. We further bound $T_1$ in the following manner:

$$T_1 \leq \frac{|g_{t,i}|}{(\sqrt{v_{t,i}}+\epsilon)(\sqrt{\beta_2 v_{t-1,i}}+\epsilon)} \times \frac{(1-\beta_2)g_{t,i}^2}{\sqrt{\beta_2 v_{t-1,i}+(1-\beta_2)g_{t,i}^2}+\sqrt{\beta_2 v_{t-1,i}}}$$

$$\leq \frac{1}{(\sqrt{v_{t,i}}+\epsilon)(\sqrt{\beta_2 v_{t-1,i}}+\epsilon)} \times \sqrt{1-\beta_2}g_{t,i}^2$$

$$\leq \frac{\sqrt{1-\beta_2}g_{t,i}^2}{(\sqrt{\beta_2 v_{t-1,i}}+\epsilon)\epsilon}.$$

Here, the third inequality is obtained by dropping $v_{t,i}$ from the denominator to obtain an upper bound. The second inequality is due to the fact that

$$\frac{|g_{t,i}|}{\sqrt{\beta_2 v_{t-1,i}+(1-\beta_2)g_{t,i}^2}} \leq \frac{1}{\sqrt{1-\beta_2}}.$$

Note that the bound of coordinates of gradient of $\ell$ automatically provides a bound on $[\nabla f(x_t)]_i$ i.e., $|[\nabla f(x_t)]_i| \leq G$ for all $i \in [d]$. Substituting the above bound on $T_1$ in Equation (3) and using the bound on $[\nabla f(x_t)]_i$, we have the following:

$$\mathbb{E}_t[f(x_{t+1})] \leq f(x_t) - \eta_t \sum_{i=1}^{d} \frac{[\nabla f(x_t)]_i^2}{\sqrt{\beta_2 v_{t-1,i}}+\epsilon} + \frac{\eta_t G\sqrt{1-\beta_2}}{\epsilon}\sum_{i=1}^{d}\mathbb{E}_t\left[\frac{g_{t,i}^2}{\sqrt{\beta_2 v_{t-1,i}}+\epsilon}\right]$$

$$+ \frac{L\eta_t^2}{2\epsilon}\sum_{i=1}^{d}\mathbb{E}_t\left[\frac{g_{t,i}^2}{\sqrt{v_{t,i}}+\epsilon}\right]$$

$$\leq f(x_t) - \eta_t \sum_{i=1}^{d} \frac{[\nabla f(x_t)]_i^2}{\sqrt{\beta_2 v_{t-1,i}}+\epsilon} + \frac{\eta_t G\sqrt{1-\beta_2}}{\epsilon}\sum_{i=1}^{d}\mathbb{E}_t\left[\frac{g_{t,i}^2}{\sqrt{\beta_2 v_{t-1,i}}+\epsilon}\right]$$

$$+ \frac{L\eta_t^2}{2\epsilon}\sum_{i=1}^{d}\mathbb{E}_t\left[\frac{g_{t,i}^2}{\sqrt{\beta_2 v_{t-1,i}}+\epsilon}\right]$$

$$\leq f(x_t) - \left(\eta_t - \frac{\eta_t G\sqrt{1-\beta_2}}{\epsilon} - \frac{L\eta_t^2}{2\epsilon}\right)\sum_{i=1}^{d}\frac{[\nabla f(x_t)]_i^2}{\sqrt{\beta_2 v_{t-1,i}}+\epsilon}$$

$$+ \left(\frac{\eta_t G\sqrt{1-\beta_2}}{\epsilon} + \frac{L\eta_t^2}{2\epsilon}\right)\sum_{i=1}^{d}\frac{\sigma_i^2}{b\sqrt{\beta_2 v_{t-1,i}}+\epsilon}.$$

The first inequality follows from the fact that $|[\nabla f(x_t)]_i| \leq G$. The third inequality follows from Lemma 1. The application of Lemma 1 is possible because $v_{t-1,i}$ is independent of random variables in $|S_t|$. The second inequality is due to the following inequality : $v_{t,i} \geq \beta_2 v_{t-1,i}$. This is obtained from the definition of $v_{t,i}$ in ADAM i.e., $v_{t,i} = \beta_2 v_{t-1,i} + (1-\beta_2)g_{t,i}^2$. From the parameters $\eta_t$, $\epsilon$ and $\beta_2$ stated in our theorem, we see that the following conditions hold: $\frac{L\eta_t}{2\epsilon} \leq \frac{1}{4}$ and

$$\frac{G\sqrt{1-\beta_2}}{\epsilon} \leq \frac{1}{4}.$$

Using these inequalities in Equation (3), we obtain

$$\mathbb{E}_t[f(x_{t+1})] \leq f(x_t) - \frac{\eta_t}{2}\sum_{i=1}^{d}\frac{[\nabla f(x_t)]_i^2}{\sqrt{\beta_2 v_{t-1,i}}+\epsilon} + \left(\frac{\eta_t G\sqrt{1-\beta_2}}{\epsilon} + \frac{L\eta_t^2}{2\epsilon}\right)\sum_{i=1}^{d}\frac{\sigma_i^2}{b(\sqrt{\beta_2 v_{t-1,i}}+\epsilon)}$$

$$\leq f(x_t) - \frac{\eta_t}{2(\sqrt{\beta_2}G+\epsilon)}\|\nabla f(x_t)\|^2 + \left(\frac{\eta_t G\sqrt{1-\beta_2}}{\epsilon^2} + \frac{L\eta_t^2}{2\epsilon^2}\right)\frac{\sigma^2}{b}$$

The second inequality follows from the fact that $0 \leq v_{t-1,i} \leq G^2$. Using telescoping sum and rearranging the inequality, we obtain

$$\frac{\eta}{2(\sqrt{\beta_2}G+\epsilon)}\sum_{t=1}^{T}\mathbb{E}\|\nabla f(x_t)\|^2 \leq f(x_1) - \mathbb{E}[f(x_{T+1})] + \left(\frac{\eta G\sqrt{1-\beta_2}}{\epsilon^2} + \frac{L\eta^2}{2\epsilon^2}\right)\frac{T\sigma^2}{b}. \quad (4)$$

Multiplying with $\frac{2(\sqrt{\beta_2}G+\epsilon)}{T\eta}$ on both sides and using the fact that $f(x^*) \le f(x_{t+1})$, we obtain the following:

$$\frac{1}{T}\sum_{t=1}^{T}\mathbb{E}\|\nabla f(x_t)\|^2 \le 2(\sqrt{\beta_2}G+\epsilon) \times \left[\frac{f(x_1)-f(x^*)}{\eta T} + \left(\frac{G\sqrt{1-\beta_2}}{\epsilon^2} + \frac{L\eta}{2\epsilon^2}\right)\frac{\sigma^2}{b}\right],$$

which gives us the desired result.

## B  Proof of Theorem 2

The proof follows along similar lines as Theorem 1 with some important differences. We, again, analyze the convergence of YOGI for general minibatch size here. Theorem 2 is obtained by setting $b = 1$. We start with the following observation:

$$f(x_{t+1}) \le f(x_t) + \langle \nabla f(x_t), x_{t+1} - x_t \rangle + \frac{L}{2}\|x_{t+1}-x_t\|^2$$

$$= f(x_t) - \eta_t \sum_{i=1}^{d}\left([\nabla f(x_t)]_i \times \frac{g_{t,i}}{\sqrt{v_{t,i}}+\epsilon}\right) + \frac{L\eta_t^2}{2}\sum_{i=1}^{d}\frac{g_{t,i}^2}{(\sqrt{v_{t,i}}+\epsilon)^2} \qquad (5)$$

The first step follows from the $L$-smoothness of the function $f$. The second step follows from the definition of YOGI update step i.e.,

$$x_{t+1,i} = x_{t,i} - \eta_t \frac{g_{t,i}}{\sqrt{v_{t,i}}+\epsilon},$$

for all $i \in [d]$. Taking the expectation at time step $t$ in Equation (2), we get the following:

$$\mathbb{E}_t[f(x_{t+1})] \le f(x_t) - \eta_t \sum_{i=1}^{d}\left([\nabla f(x_t)]_i \times \mathbb{E}_t\left[\frac{g_{t,i}}{\sqrt{v_{t,i}}+\epsilon}\right]\right) + \frac{L\eta_t^2}{2}\sum_{i=1}^{d}\mathbb{E}_t\left[\frac{g_{t,i}^2}{(\sqrt{v_{t,i}}+\epsilon)^2}\right]$$

$$= f(x_t) - \eta_t \sum_{i=1}^{d}\left([\nabla f(x_t)]_i \times \mathbb{E}_t\left[\frac{g_{t,i}}{\sqrt{v_{t,i}}+\epsilon} - \frac{g_{t,i}}{\sqrt{v_{t-1,i}}+\epsilon} + \frac{g_{t,i}}{\sqrt{v_{t-1,i}}+\epsilon}\right]\right)$$

$$+ \frac{L\eta_t^2}{2}\sum_{i=1}^{d}\mathbb{E}_t\left[\frac{g_{t,i}^2}{(\sqrt{v_{t,i}}+\epsilon)^2}\right]$$

$$\le f(x_t) - \eta_t \sum_{i=1}^{d}\frac{[\nabla f(x_t)]_i^2}{\sqrt{v_{t-1,i}}+\epsilon} + \eta_t \sum_{i=1}^{d}|[\nabla f(x_t)]_i|\left|\mathbb{E}_t\underbrace{\left[\frac{g_{t,i}}{\sqrt{v_{t,i}}+\epsilon} - \frac{g_{t,i}}{\sqrt{v_{t-1,i}}+\epsilon}\right]}_{T_1}\right|$$

$$+ \underbrace{\frac{L\eta_t^2}{2}\sum_{i=1}^{d}\mathbb{E}_t\left[\frac{g_{t,i}^2}{(\sqrt{v_{t,i}}+\epsilon)^2}\right]}_{T_2}. \qquad (6)$$

The second equality follows from the fact that $g_t$ is an unbiased estimate of $\nabla f(x_t)$ i.e., $\mathbb{E}[g_t] = \nabla f(x_t)$. The key difference here in comparison to proof of Theorem 1 is that the deviation to bound in $T_1$ is from $\frac{g_{t,i}}{\sqrt{v_{t-1,i}}+\epsilon}$ as opposed to $\frac{g_{t,i}}{\sqrt{\beta_2 v_{t-1,i}}+\epsilon}$ in proof of ADAM. Our aim is to bound the terms $T_1$ and $T_2$ in the above inequality. We bound the term $T_1$ in the following manner:

$$T_1 \le |g_{t,i}|\left|\frac{1}{\sqrt{v_{t,i}}+\epsilon} - \frac{1}{\sqrt{v_{t-1,i}}+\epsilon}\right|$$

$$= \frac{|g_{t,i}|}{(\sqrt{v_{t,i}}+\epsilon)(\sqrt{v_{t-1,i}}+\epsilon)}\left|\frac{v_{t,i}-v_{t-1,i}}{\sqrt{v_{t,i}}+\sqrt{v_{t-1,i}}}\right|$$

$$= \frac{|g_{t,i}|}{(\sqrt{v_{t,i}}+\epsilon)(\sqrt{v_{t-1,i}}+\epsilon)} \times \frac{(1-\beta_2)g_{t,i}^2}{\sqrt{v_{t,i}}+\sqrt{v_{t-1,i}}} \le \frac{\sqrt{1-\beta_2}g_{t,i}^2}{(\sqrt{v_{t-1,i}}+\epsilon)\epsilon}.$$

The second equality is from the update rule of YOGI which is $v_{t,i} = v_{t-1,i} - (1-\beta_2)\text{sign}(v_{t-1,i} - g_{t,i}^2)g_{t,i}^2$. The last inequality is due to the fact that

$$\frac{|g_{t,i}|}{\sqrt{v_{t,i}} + \sqrt{v_{t-1,i}}} \leq \frac{1}{\sqrt{1-\beta_2}}.$$

The above inequality in turn follows from the fact that either $\frac{|g_{t,i}|}{\sqrt{v_{t-1,i}}} \leq 1$ when $v_{t-1,i} \geq g_{t,i}^2$ or $\frac{|g_{t,i}|}{\sqrt{v_{t,i}}} \leq \frac{1}{\sqrt{1-\beta_2}}$ when $v_{t-1,i} < g_{t,i}^2$. We next bound the term $T_2$ as follows:

$$T_2 = \frac{L\eta_t^2}{2}\sum_{i=1}^{d}\mathbb{E}_t\left[\frac{g_{t,i}^2}{(\sqrt{v_{t,i}}+\epsilon)^2}\right] \leq \frac{L\eta_t^2}{2\epsilon\sqrt{\beta_2}}\sum_{i=1}^{d}\mathbb{E}_t\left[\frac{g_{t,i}^2}{\sqrt{v_{t-1,i}}+\epsilon}\right].$$

The inequality is due to the following : $v_{t,i} \geq \beta_2 v_{t-1,i}$. To see this, first note that $v_{t,i} = v_{t-1,i} - (1-\beta_2)\text{sign}(v_{t-1,i} - g_{t,i}^2)g_{t,i}^2$. If $v_{t-1,i} \leq g_{t,i}^2$, then it is easy to see that $v_{t,i} \geq v_{t-1,i}$. Consider the case where $v_{t-1,i} > g_{t,i}^2$, then we have

$$v_{t,i} = v_{t-1,i} - (1-\beta_2)g_{t,i}^2 \geq \beta_2 v_{t-1,i}.$$

Therefore, $v_{t,i} \geq \beta_2 v_{t-1,i}$. Substituting the above bounds on $T_1$ and $T_2$ in Equation (6), we obtain the following bound:

$$\mathbb{E}_t[f(x_{t+1})] \leq f(x_t) - \eta_t \sum_{i=1}^{d}\frac{[\nabla f(x_t)]_i^2}{\sqrt{v_{t-1,i}}+\epsilon} + \frac{\eta_t G\sqrt{1-\beta_2}}{\epsilon}\sum_{i=1}^{d}\mathbb{E}_t\left[\frac{g_{t,i}^2}{\sqrt{v_{t-1,i}}+\epsilon}\right]$$

$$+ \frac{L\eta_t^2}{2\epsilon\sqrt{\beta_2}}\sum_{i=1}^{d}\mathbb{E}_t\left[\frac{g_{t,i}^2}{\sqrt{v_{t-1,i}}+\epsilon}\right]$$

$$\leq f(x_t) - \left(\eta_t - \frac{\eta_t G\sqrt{1-\beta_2}}{\epsilon} - \frac{L\eta_t^2}{2\epsilon\sqrt{\beta_2}}\right)\sum_{i=1}^{d}\frac{[\nabla f(x_t)]_i^2}{\sqrt{v_{t-1,i}}+\epsilon}$$

$$+ \left(\frac{\eta_t G^2(1-\beta_2)}{2\epsilon} + \frac{L\eta_t^2}{2\epsilon\sqrt{\beta_2}}\right)\sum_{i=1}^{d}\frac{\sigma_i^2}{b\sqrt{v_{t-1,i}}+\epsilon}.$$

The first inequality follows from the fact that $|[\nabla f(x_t)]_i| \leq G$. The second inequality follows from Lemma 1. Now, from our theorem result, we observe that,

$$\frac{G\sqrt{1-\beta_2}}{\epsilon} \leq \frac{1}{4},$$
$$\frac{L\eta_t}{2\epsilon\sqrt{\beta_2}} \leq \frac{1}{4}.$$

Using these inequalities in Equation (6), we obtain

$$\mathbb{E}_t[f(x_{t+1})] \leq f(x_t) - \frac{\eta_t}{2}\sum_{i=1}^{d}\frac{[\nabla f(x_t)]_i^2}{\sqrt{v_{t-1,i}}+\epsilon} + \left(\frac{\eta_t G\sqrt{1-\beta_2}}{\epsilon} + \frac{L\eta_t^2}{2\epsilon\sqrt{\beta_2}}\right)\sum_{i=1}^{d}\frac{\sigma_i^2}{b\sqrt{v_{t-1,i}}+\epsilon}$$

$$\leq f(x_t) - \frac{\eta_t}{2(\sqrt{2}G+\epsilon)}\|\nabla f(x_t)\|^2 + \left(\frac{\eta_t G\sqrt{1-\beta_2}}{\epsilon^2} + \frac{L\eta_t^2}{2\epsilon^2\sqrt{\beta_2}}\right)\frac{\sigma^2}{b}$$

The second inequality follows from the fact that $0 \leq v_{t-1,i} \leq 2G^2$. Using telescoping sum and rearranging the inequality, we obtain

$$\frac{\eta}{2(\sqrt{2}G+\epsilon)}\sum_{t=1}^{T}\mathbb{E}\|\nabla f(x_t)\|^2 \leq f(x_1) - \mathbb{E}[f(x_{T+1})] + \left(\frac{\eta G\sqrt{1-\beta_2}}{\epsilon^2} + \frac{L\eta^2}{2\epsilon^2\sqrt{\beta_2}}\right)\frac{T\sigma^2}{b}. \quad (7)$$

Multiplying with $\frac{2(\sqrt{2}G+\epsilon)}{\eta}$ on both sides and using the fact that $f(x^*) \leq f(x_{t+1})$ gives us the desired result.

# C  Auxiliary Lemma

The following result is useful for bounding the variance of the updates of the algorithms in this paper.

**Lemma 1.** *For the iterates $x_t$ where $t \in [T]$ in Algorithm 1 and 2, the following inequality holds:*

$$\mathbb{E}_t[\|g_{t,i}\|^2] \leq \frac{\sigma_i^2}{b} + [\nabla f(x_t)]_i^2,$$

*for all $i \in [d]$.*

*Proof.* Let us define the following notation for the ease of exposition:

$$\zeta_t = \frac{1}{|S_t|} \sum_{s \in S_t} \left([\nabla \ell(x_t, s)]_i - [\nabla f(x_t)]_i\right).$$

Using this notation, we obtain the following bound:

$$
\begin{aligned}
\mathbb{E}_t[g_{t,i}^2] &= \mathbb{E}_t[\|\zeta_t + \nabla f(x_t)\|^2] \\
&= \mathbb{E}_t[\zeta_t^2] + [\nabla f(x_t)]_i^2 \\
&= \frac{1}{b^2} \mathbb{E}_t\left[\left(\sum_{s \in S_t} \left([\nabla \ell(x_t, s)]_i - [\nabla f(x_t)]_i\right)\right)^2\right] + [\nabla f(x_t)]_i^2 \\
&= \frac{1}{b^2} \mathbb{E}_t\left[\sum_{s \in S_t} \left([\nabla \ell(x_t, s)]_i - [\nabla f(x_t)]_i\right)^2\right] + [\nabla f(x_t)]_i^2 \\
&\leq \frac{\sigma_i^2}{b} + [\nabla f(x_t)]_i^2.
\end{aligned}
$$

The second equality is due to the fact that $\zeta_t$ is a mean 0 random variable. The third equality follows from Lemma 2. The last inequality is due to the fact that $\mathbb{E}_{s \sim \mathbb{P}}[([\nabla \ell(x_t, s)]_i - [\nabla f(x_t)]_i)^2] \leq \sigma_i^2$ for all $x \in \mathbb{R}^d$. $\qquad \square$

**Lemma 2.** *For random variables $z_1, \ldots, z_r$ are independent and mean 0, we have*

$$\mathbb{E}\left[\|z_1 + \ldots + z_r\|^2\right] = \mathbb{E}\left[\|z_1\|^2 + \ldots + \|z_r\|^2\right].$$

*Proof.* We have the following:

$$\mathbb{E}\left[\|z_1 + \ldots + z_r\|^2\right] = \sum_{i,j=1}^{r} \mathbb{E}[z_i z_j] = \mathbb{E}\left[\|z_1\|^2 + \ldots + \|z_r\|^2\right].$$

The second equality follows from the fact that $z_i$'s are independent and mean 0. $\qquad \square$

# D More Experiment Results

Table 5: Train and test loss comparison for Deep AutoEncoders. Standard errors with $2\sigma$ are shown over 6 runs are shown. All our experiments were run for 5000 epochs utilizing the ReduceLRonPlateau schedule with patience of 20 epochs and decay factor of 0.5 with a batch size of 128. Also we report numbers from prior work for reference, but their experimental setup (batch-size, learning rate, etc) are different.

| Method | LR | $\beta_1$ | $\beta_2$ | $\epsilon$ | CURVES | |
|---|---|---|---|---|---|---|
| | | | | | Train Loss | Test Loss |
| PT + NCG [20] | - | - | - | - | 0.74 | 0.82 |
| RAND+HF [20] | - | - | - | - | 0.11 | 0.20 |
| PT + HF [20] | - | - | - | - | 0.10 | 0.21 |
| KSD [36] | - | - | - | - | 0.17 | 0.25 |
| HF [36] | - | - | - | - | 0.13 | 0.19 |
| ADAM (Default) | $10^{-3}$ | 0.9 | 0.999 | $10^{-8}$ | $0.09 \pm 0.16$ | $0.16 \pm 0.02$ |
| ADAM (Modified) | $10^{-3}$ | 0.9 | 0.999 | $10^{-3}$ | $0.12 \pm 0.17$ | $0.17 \pm 0.01$ |
| YOGI (Ours) | $10^{-2}$ | 0.9 | 0.9 | $10^{-3}$ | $0.11 \pm 0.01$ | $\mathbf{0.15 \pm 0.01}$ |
| YOGI (Ours) | $10^{-2}$ | 0.9 | 0.999 | $10^{-3}$ | $0.20 \pm 0.01$ | $0.25 \pm 0.02$ |

Table 6: Test accuracy for DeepSets on ModelNet40. Standard errors with $2\sigma$ are shown over 6 runs are shown. All our experiments were run for 500 epochs utilizing the ReduceLRonPlateau schedule with patience of 20 epochs and decay factor of 0.5 with a batch size of 128. Also we report numbers from original paper for reference, which employs a highly tuned learning rate schedule.

| Method | LR | $\beta_1$ | $\beta_2$ | $\epsilon$ | Test Accuracy (%) |
|---|---|---|---|---|---|
| Adam [39] | - | - | - | - | $87.0 \pm 1.0$ |
| ADAM (Default) | $10^{-3}$ | 0.9 | 0.999 | $10^{-8}$ | $87.71 \pm 0.25$ |
| ADAM (Modified) | $10^{-3}$ | 0.9 | 0.999 | $10^{-3}$ | $88.41 \pm 0.33$ |
| YOGI (Ours) | $10^{-2}$ | 0.9 | 0.999 | $10^{-3}$ | $87.65 \pm 0.15$ |
| YOGI (Ours) | $5 \times 10^{-3}$ | 0.9 | 0.999 | $10^{-3}$ | $\mathbf{88.73 \pm 0.28}$ |

Table 7: Test F1 score for Named Entity Recognition task using CNN-LSTM-CRF model on BC5CDR bio-medical dataset. Standard errors with $2\sigma$ calculated over 6 runs are shown. All our experiments were run for 50 epochs utilizing the ReduceLRonPlateau schedule with patience of 10 epochs and decay factor of 0.5 with a batch size of 2000 words. We also report performance score from one the best performing approaches for reference, which employs a highly tuned learning rate schedule.

| Method | LR | $\beta_1$ | $\beta_2$ | $\epsilon$ | Test F1 (%) |
|---|---|---|---|---|---|
| SGD [37] | - | - | - | - | 88.78 |
| ADAM (Default) | $10^{-3}$ | 0.9 | 0.999 | $10^{-8}$ | $88.75 \pm 0.23$ |
| ADAM (Modified) | $10^{-3}$ | 0.9 | 0.999 | $10^{-3}$ | $88.86 \pm 0.22$ |
| YOGI (Ours) | $10^{-2}$ | 0.9 | 0.999 | $10^{-3}$ | $\mathbf{89.20 \pm 0.17}$ |