[Reviews · NeurIPS 2018]

Reviewer 1



## Summary: The authors analyze an existing adaptive stochastic optimization method, ADAM, and propose/analyze a new adaptive stochastic optimization method, YOGI, for solving non-convex stochastic optimization problems. Bounds are given for the expected gradient of an ergodic average of the iterates produced by the algorithms applied to an L-smooth function, and these bounds converge to zero with time. The authors give several numerical results showing that their algorithm has state-of-the-art performance for different problems. In addition, they achieve this performance with little tuning, unlike in the classical SGD. A motivation behind their work is a paper [27] that shows that a recent adaptive algorithm, ADAM, can fail to converge even for simple convex problems, when the batch size is kept fix. Their analysis points to the idea of using growing batch sizes. An idea which they show does translates into good performance. ## Quality: Over all good. The definition of SFO complexity depends on the definition of $\delta$-accuracy for a solution $x$, which was defined as $x$ satisfying $\|\nabla f(x)\|^2 < \delta$. However, the theorems only prove bounds on the expected $\|\nabla f(x_a)\|^2$ where $a$ is uniformly sampled in [T] and $x_a$ is a random object that depends on $\{s_t\}$ sampled from $\mathbb{P}$. It would be good to say a few words about how, given a sequence $\{x_a\}$ that satisfies the bounds in the theorems, we can find *one* solution that is $\delta$-accurate without much computational effort. If the expected value of the squared norm of the gradient of f at x_a is <= O (1/ T) doesn't this imply that the complexity to get an error of 1/T is T ? Where does the $^2$ in $O(1/\delta^2)$ come from? What am I missing? I would suggest that in the expectation in Th1. the authors are explicit that the expectation is being taken with respect with to $a$ and $\{s_t\}$ sampled from $\mathbb{P}$. The same goes with the expectations in Sec. 2. I would suggest that in Alg. 1, the authors write the v_t update equation in the form v_t = v_{t-1} - (1 - \beta_2)(v_{t-1} - g_t^2) g_t^2, to facilitate comparing YOGI with ADAM. ## Clarity: The paper is very well written and easy to understand. Great job ! ## Originality: The algorithm, even if a simple modification of ADAM, is new, and the proofs seem to be new as well. ## Significance: Empirical performance gains are not consistently amazing, but the fact that their algorithm requires little tuning is a great thing. It is push in the right direction. SGD sometimes outperforms YOGI. Is it the case than if we invest an equal effort in tuning YOGI, maybe after introducing some extra parameters into it, that YOGI can beat SGD in these cases as well? It is rather unfortunate that the authors do not include any ideas behind the proofs in the main text, and have the reviewer read 6 extra pages of proofs. There is a myriad of paper that show method XYZ gives better empirical performance on different ML than the previous state-of-the-art. What distinguishes one paper from the other, is the support from theory. In this case, I would suggest passing many of the details about the different numerical experiments to the appendix, and bringing some "juice" to the main document. In particular, the authors should emphasize what are the key technical steps in their proof techniques that distinguish their work from previous analysis of similar algorithms. This prevents the reader to appreciate the full significance and originality of their work. It would be good if the authors could write in the paper if they will, or not, release code after and if the paper gets published. A written commitment to reproducibility, no links, would be enough.

Reviewer 2



Summary: This paper provides new theoretical insights for non-convex problems for the optimizer Adam and Yogi (the proposed optimizer). In particular, the non-convergence of Adam is tackled with increasing batch size. Yogi provides more control than Adam in tuning the effective learning rate. The authors also present experiments comparing Yogi with Adam. The paper is very clear in terms of writing. Yogi idea is original but not sufficiently motivated. The work presented is not very significant due to certain weaknesses (see below). Strengths: 1) Theoretical results for non-convex problems for Adam (actually RMSProp) and Yogi are interesting. 2) Many experiments were presented though not sufficient (can be biased due to hyperparameter selection). Weaknesses: 1) Considering g_t only rather than m_t in update step (in Appendix) makes the update step more similar to RMSProp, actually Adam will be exactly the same as RMSProp when beta_1 = 0 (if one excludes debiasing step). So there is no sync between theory and experiments. I would prefer the algorithm being called RMSProp rather than Adam in theoretical results. In discussion, the authors mention that they set beta_1 = 0.9 inspired from theoretical analysis, but in theory they set beta_1=0. This needs to be clarified and should be consistent. 2) No proper motivation for the update step, the explanation for the update step is very minor. I think it would be better if the authors dwell a bit more into the details what makes the algorithm special for instance explain different cases especially for different cases of v_{t-1} and g_{t}^2 where Adam fails and Yogi does it better. Since, there is only a minor modification in the update step, as compared to RMSprop, this is a critical issue. The formula itself doesn't look like a natural choice, thus it needs to be justified. 3) In theoretical results (both in Adam and Yogi), there is O(G^2/\epsilon^2) term as coefficient for variance, which can be very huge if epsilon is small. And usually epsilon is very small. It would be nice to see if one can decrease the dependence on epsilon. Regarding this aspect there is no improvement compared to the literature. 4) Weak experiments section, there is no proper rule in the choice of learning rate and \epsilon. I think one must choose the tuned hyperparameters from a set (need not be very big). One cannot be sure if the performance is due to hyperparameter selection or actually due to performance gains. Improvements: 1) What about debiasing results? At least brief comments on the changes in the theory. 2) Some theoretical insights when beta_1 \neq 0 for Adam and Yogi. 3) Relation to Adagrad, if v_{t-1}

Reviewer 3



This paper proposed convergence analysis for ADAM under certain parameter settings and proved the convergence to a stationary point for nonconvex problems. Moreover, the authors proposed an adaptive SGD (i.e., YOGI) and evaluated it on the different learning models. Generally, the studied problem is interesting and the proposed algorithm demonstrated good performance on its testing problems. However, some important issues are not clearly stated in the current manuscript: 1. This paper has stated that YOGI controls the increase in effective learning rate, but it is not clear how it impacts the learning rate? 2. The proof of the convergence in ADAM only considered the case that $\beta_1=0$. Thus it is also not very clear how about the convergence proof with the condition $\beta_1\in(0,1)$? 3. There are some types and improper presentations in Appendix. For example, in the first line of page 12, the update rule $v_t,i$ of YOGI does not match with the update rule in Algorithm 2.

Reviewer 4



This paper studies the adaptive gradient methods (e.g, ADAM) for nonconvex stochastic problems. Specifically, the authors prove the convergence to the stationary point using the increasing batch size. They also propose a new algorithm of YOGI and achieve the same convergence result. I have some concers on the "increaing minibatch size". 1. The authors claim that increasing batch size leads to convergence. It may be ambiguous. From Corollaries 2,3,4,5, we can see that the authors use a large and fixed batch size in the order of O(T), where T is the number of iterations. In my opinions, increasing batch size means that the batch size is increaing during the iterations and it may be more practical than the fixed and large batch size. Of course, it may be more chanllenging to prove the convergence with the real increaing batch size. Comment after author rebuttal: the authors said that " the same theoretical guarantee can be obtained for increasing minibatch b_t = Θ(t)". I am not convinced of it. Hope the authors to include a detailed proof for the case of b_t = Θ(t) in the final version. If the analysis only applies to the case of b_t = Θ(T), please point it clearly and do not mislead the readers. 2. More seriously, the theories in this paper require the batch size to be $b=O(T)$. In practice, we may run the algorithm for several epoches, e.g., 100 epoches. Let N be the sample size, then T=O(100N). However, the batchsize we ofteh use is much smaller than N, e.g., b=100, in this case, b=O(T) is an unpractical requirement. I think this issue is not due to the weakness of ADAM, but the weakness of the proof in this paper. Comment after author rebuttal: I have made a mistake in the previous comment, that is, T should be O(100N/b) and b=O(T) leads to b=O(10\sqrt{N}). This setting is acceptable in practice. From Table 1 in "Neon2: Finding Local Minima via First-Order Oracles", we know SGD needs O(1/\delta^4) iterations to find x such that $ ||\nabla f(x)|| \leq detla$. The complexity proved in Corollary 3 is O(1/\delta^2) such that $ ||\nabla f(x)|| \leq detla$. I do not think ADAM has a faster theoretical convergence rate than SGD. Thus it may verify that this paper has made some unreasonable assumptions. Comment after author rebuttal: The authors persuaded me. Since the minibatch is O(T) and SFO complexity measures the total number of stochastic oracle calls. So the final compleixty of ADAM is O(T^2) = O(1/\delta^4). Hope the authors to make it clear in the final version and do not mislead the readers that ADAM is theoretically faster than SGD (they have the same theoretical complexity in the convex case). Corollary 1 in this paper is also ambiguous. I have read the introduction of [8]. [8] proved that their method needs O(1/epsilon^2) iterations to find x such that $ ||\nabla f(x)||^2 \leq epsilon$. However, from Corollary 2, we know ADAM needs O(1/epsilon) iterations to find x such that $ ||\nabla f(x)||^2 \leq epsilon$. Thus, the comparison in Section: Discussion about Theoretical Results is not correct. Comment after author rebuttal: The authors persuaded me. Please see the above comment. In conclusion, the study in this paper may be significative in practice. But the theoretical analysis is not satisfied. Final comment after author rebuttal: I am pleased with the theoretical analysis now. I do not evalute the experiments due to lack of experience.